# Sleep Problems, Social Anxiety and Stuttering Severity in Adults Who Do and Adults Who Do Not Stutter

**DOI:** 10.3390/jcm12010161

**Published:** 2022-12-25

**Authors:** Hiwa Mohammadi, Soroush Maazinezhad, Elaheh Lorestani, Ali Zakiei, Kenneth M. Dürsteler, Annette Beatrix Brühl, Dena Sadeghi-Bahmani, Serge Brand

**Affiliations:** 1Neuroscience Research Center, Health Technology Institute, Kermanshah University of Medical Sciences, Kermanshah 6734667149, Iran; 2Sleep Disorders Research Center, Health Institute, Kermanshah University of Medical Sciences, Kermanshah 6734667149, Iran; 3Department of Psychology, Tarbiat Modares University, Tehran 1411713116, Iran; 4Department of Addictive Disorders, Psychiatric University Clinic Basel, 4002 Basel, Switzerland; 5Department for Psychiatry, Psychotherapy and Psychosomatic, Psychiatric Hospital, University of Zurich, 3002 Zurich, Switzerland; 6Center for Affective, Stress and Sleep Disorders (ZASS), Psychiatric University Hospital Basel, 4002 Basel, Switzerland; 7Department of Psychology, Stanford University, Stanford, CA 94305, USA; 8Division of Sport and Psychosocial Health, Department of Sport, Exercise, and Health, University of Basel, 4052 Basel, Switzerland; 9Substance Abuse Prevention Research Center and Sleep Disorders Research Center, Kermanshah University of Medical Sciences, Kermanshah 6715847141, Iran; 10School of Medicine, Tehran University of Medical Sciences (TUMS), Tehran 1417466191, Iran; 11Center for Disaster Psychiatry and Disaster Psychology, Psychiatric University Hospital Basel, 4002 Basel, Switzerland

**Keywords:** adults, stuttering, sleep quality, sleep disturbances, social anxiety

## Abstract

**Background:** While there is sufficient evidence that children and adolescents who stutter reported more impaired sleep compared to children and adolescents who did not stutter, findings among adults who stutter (AWS) were scarce. Furthermore, stuttering is associated with issues related to verbal communication in a social context. As such, it was conceivable that AWS reported higher scores for social anxiety, compared to adults who do not stutter (AWNS). In the present study, we tested whether AWS reported higher sleep complaints compared to AWNS. We further tested whether scores for social anxiety and stuttering independently predicted sleep disturbances. **Methods:** A total of 110 AWS (mean age; 28.25 years, 27.30% females) and 162 AWNS (mean age; 29.40 years, 51.20% females) completed a series of self-rating questionnaires covering sociodemographic information, sleep disturbances and social anxiety. Adults with stuttering further completed a questionnaire on stuttering. **Results:** Compared to AWNS, AWS reported a shorter sleep duration, a lower sleep efficiency, higher scores for drug use in terms of sleep-promoting medications (significant *p*-values and medium effect sizes), and an overall higher PSQI score (significant *p*-values and large effect size), when controlling for age and social anxiety. Next, while *p*-values were always significant for subjective sleep quality, sleep disturbances, and daytime functioning, when controlling for age and social anxiety, their effect sizes were trivial or small. For sleep latency, the *p*-value was not significant and the effect size was trivial. Among AWS, higher scores for stuttering and older age, but not social anxiety, predicted higher sleep disturbances. The association between higher sleep disturbances and higher stuttering severity was greatest among those AWS with highest scores for social anxiety. **Conclusions:** When compared to AWNS, AWS self-reported higher sleep disturbances, which were associated with older age, and higher scores for stuttering severity, but not with social anxiety. Adults who stutter might be routinely asked for their sleep quality.

## 1. Introduction

About five out of 100 children and adolescents and about one out of 100 adults are reported to suffer from developmental stuttering. Developmental stuttering is considered a dysfunctional speech motor control condition [1]. Among adults, stuttering is related to lower wellbeing and quality of life [2,3], and to a lower income [4]. Furthermore, given that stuttering impacts speech and communication, individuals who stutter also report more issues in a social context [5].

### 1.1. Symptoms of Stuttering and Its Psychosocial Consequences

Stuttering is not a uniform phenomenon, but a dynamic and dysfunctional speech motor condition which varies in frequency, severity, traits of dysfluencies, social context, social expectancies, speech duration, the subjectively perceived importance of social interactions, and the subjective satisfaction with speech performance [6,7,8].

While the neurological mechanisms of stuttering remains elusive, current therapies focus on behavioral modification of speech such as fluency shaping strategies [9,10] and on the adjustment of the long-lasting psychological consequences of stuttering [11,12]. In addition, given that stuttering may increase as a function of a person’s subjective importance of a social interaction and its related social anxiety, social anxiety management appeared to be a promising intervention for a better management of the disorder in adults who stutter (AWS) [13].

As regards the association between stuttering severity and social anxiety, there is sufficient evidence that stuttering severity and social anxiety go in tandem [5,14,15,16,17,18,19,20,21,22,23]. To illustrate, Yang, Jia, Siok and Tan [23] employed fMRI techniques to compare the neuronal processes of functional connectivity among adults who stutter (AWS; n = 19) and adults with no stuttering (AWNS; n = 19). Yang et al. [23] showed that compared to AWNS, AWS showed a higher functional connectivity, and thus a more severe processing among those networks highly involved in anxiety processing. Thus, this study identified a higher anxiety-related neuronal activity among AWS than AWNS. Chu et al. [5] compared scores for self-rated social anxiety among Japanese AWS (n = 130; mean age = 41.5 years; 111 males) and Japanese AWNS (n = 114; mean age = 39.5 years; 53 males). As expected, and in line with international results, AWS reported statistically significantly higher social anxiety scores than AWNS. Iverach et al. [16] assessed a total of 275 AWS (age range: 10–80 years; 79.6% males) and compared psychological issues and speech performance among AWS with diagnosed social anxiety disorder (n = 82; 29.8%) and without social anxiety disorder (n = 193; 70.2%). Interestingly, when compared to AWS without social anxiety disorder, AWS with diagnosed social anxiety disorder did not report higher scores for stuttering severity or percentage of syllables stuttered, while AWS with diagnosed social anxiety disorder reported more speech dissatisfaction and avoidance of speaking situations, along with significantly more psychological issues, and a more negative view of their stuttering and speech on everyday life. Given this, it appears that not the stuttering severity per se, but the cognitive-emotional elaboration of possible consequences of stuttering in a social context appeared to be the driving factor of social anxiety. In this view, Tomisato et al. [21] showed that the favorable coping style of a person with stuttering mitigated the severity of social anxiety.

To summarize, compared to AWNS, AWS scored higher on dimensions of social anxiety; such self-rated social anxiety was associated with an increased neural connectivity related to social anxiety perception and anxiety elaboration. Furthermore, AWS scoring high on social anxiety understood a specific social context as more uncomfortable.

### 1.2. The Importance of Sleep for Emotion Regulation

There is evidence that sufficient and restorative sleep is essential for accurate and precise cognitive, emotional, behavioral and interactional processes [24,25,26,27,28]. Good sleep and a normal circadian rhythm are closely related to mental well-being and normal brain function [29]. More specifically, sufficient and restoring sleep is associated with a higher performance of emotion regulation [30,31,32,33,34,35,36].

As regards sleep quality and anxiety in general, and social anxiety specifically, a higher sleep quality impairment was associated with higher scores for social anxiety [37,38,39,40,41]. As such, it appears plausible that adult individuals who stutter and report signs of social anxiety might also report more impaired sleep (see below).

### 1.3. Sleep Patterns among Individuals with Stuttering

Given this background, it is not surprising, compared that to individuals with no stuttering, individuals with stuttering reported more impaired sleep. Such an association was already observable among children with stuttering: Children with stuttering scored high on nightmares, sleep deprivation, and insomnia near the age of onset of stuttering. Along these lines, recent studies reported a significant association between developmental stuttering and sleep problems among children and adolescents [42,43]. Children and adolescents who stutter self-reported significantly higher sleep problems than those with no stuttering [44], and compared to adolescents and young adults with no stuttering, adolescents and young adults who stutter were reported to sleep an average of 20 min less per night. Additionally, 15% of those children and adolescents who stutter reported difficulties falling or staying asleep almost every day or every day, which was twice as likely as children and adolescents who did not stutter [43]. Thus, while there is increasing evidence of the association between poor sleep and stuttering among children and adolescents, interestingly, such data appear to be fully missing for adults who stutter (AWS).

### 1.4. The Present Study

Given the background mentioned above, the first aim of the present study was to assess subjective sleep patterns among adults with stutter compared to a sample of the general population with no stuttering. The second aim of the study was to associate self-rated sleep quality with dimensions of stuttering severity. And the third aim of the present study was to associate self-rated sleep quality with dimensions of stuttering severity, while also taking social anxiety into consideration.

The following three hypotheses and two research questions were formulated. First, based on results from children and adolescents with stuttering [42,43,44], we expected that AWS would report more sleep complaints than AWNS. Second, based on the results not related to stuttering [37,38,39,40,41], we expected that poor self-rated sleep would be associated with higher scores for social anxiety. Third, given that stuttering was associated with social anxiety [5,14,15,16,17,18,19,20,21,22,23], and given that poor sleep is associated with higher anxiety scores [37,38,39,40,41], we expected that higher scores for stuttering severity would be associated with higher scores for sleep disturbances.

With the first research question, we investigated among AWS if and to what extent correlation coefficients between sleep disturbances and stuttering severity changed, if social anxiety was introduced as co-variate, and if and to what extent correlation coefficients between sleep disturbances and social anxiety changed if stuttering severity was introduced as a co-variate.

With the second research question, we investigated which dimensions (age, social anxiety, stuttering) predicted sleep disturbances among AWS.

We claim that the present results have the potential to investigate the intertwined associations between stuttering severity, subjective sleep, and social anxiety so as to allow for possible interventions.

## 2. Materials and Methods

### 2.1. Participants

In this case-control study, 110 AWS and 162 AWNS with the age range of 18 to 60 years were enrolled. AWS were recruited from the Iranian stuttering association. This association is a support group and not an experts’ association. AWNS were recruited from the general population.

For AWS, inclusion criteria were: (1) age 18 years and older; (2) stuttering, as ascertained by the participants and as confirmed by a speech-language pathologist; (3) willing and able to complete a series of self-rating questionnaires (see below) on sociodemographic information, stuttering, social anxiety, and subjective sleep patterns; and (4) signed written informed consent. Exclusion criteria were: (1) psychiatric issues as ascertained by an experienced clinical psychologist and based on a brief psychiatric interview for DSM-5 psychiatric disorders [45]; (2) neurological or immunological disorders within the last 12 months; (3) current intake of sleep- and mood-altering medications such as mood stabilizers, antidepressants, benzodiazepines, antihypertensives, or nootropics; and (4) women who are pregnant or breastfeeding, as both conditions may alter current sleep and mood conditions.

For AWNS, inclusion and exclusion criteria were identical to those of AWS, though with the explicit exclusion of any sings of stuttering.

A total of 117 AWS were approached, and 110 (94%) agreed to participate at the study. Further, a total of 168 AWNS were approached, and 162 (96%) agreed to participate in the study.

### 2.2. Procedure

Between April and August 2021, individuals with and with no stuttering were approached to participate in the representative cross-sectional and questionnaire-based study. They were fully informed about the aims of the study and the secure data handling. Once they had signed the written informed consent, participants completed a series of self-rating questionnaires available on the website of the Kermanshah University of Medical Sciences (KUMS). The study was advertised on SNS (social network sites) for AWS and AWNS. Both AWS and AWNS completed the same questionnaires, though AWS also completed a specific questionnaire on stuttering (see below). Participants could contact the staff involved in the present study in case of questions, and all information was kept strictly confidential.

### 2.3. Measures

#### 2.3.1. Sociodemographic Information

Participants reported on their age (years), gender at birth (male; female), civil status (single, married, divorced), current job position (employed; unemployed; independent worker), socio-economic status (low, moderate, or high income), and highest educational degree (high school; undergraduate; postgraduate).

#### 2.3.2. Subjective Sleep Quality

To assess subjective sleep quality over the last four weeks, participants completed the Farsi version [34,46,47] of the Pittsburgh Sleep Quality Index (PSQI; [48]. The questionnaire has seven sub-scales including subjective sleep quality, sleep latency, sleep duration, habitual sleep efficiency, sleep disturbances, use of sleep medication, and daytime dysfunction. The total PSQI index is calculated based on seven sub-scale scores and a total score between 0 and 21 is obtained. The higher the score, the lower the sleep quality. A score of 6 is a cut point for disordered sleep quality [47,49,50,51,52]. The current Cronbach’s alpha was 0.77.

#### 2.3.3. Social Avoidance and Distress

To assess social avoidance and social distress, participants completed the Farsi version [53] of the Social Avoidance and Distress Scale (SADS) [54]. The SADS is a self-rating scale to assess social anxiety, and consists of 28 true/false items that measure distress, discomfort, fear, and avoidance; a higher sum score reflects higher social anxiety or distress. In general, individuals scoring high on the SADS reported lower scores for self-esteem, lower values of self-confidence, and higher scores for need for affiliation, need for change, and need for dominance [55]. Furthermore, to estimate whether and to which degree social anxiety moderated the associations between stuttering severity and sleep disturbances, scores for social anxiety were split in tertiles with the following cut-off values: no social anxiety: m = 9.52 (SD = 1.52); moderate social anxiety: m = 13.56 (SD = 1.16); high social anxiety: m = 18.25 (SD = 1.71).

#### 2.3.4. Self-Rated Stuttering

Individuals who stutter completed the Farsi version [56] of the Subjective Screening of Stuttering (SSS) [57]. The SSS is a self-reported instrument for evaluating perceived stuttering severity, the level of internal or external locus of control, and reported word or situation avoidance. The questionnaire consists of eight items; two measure severity (e.g., “How would you score your speech with the following audiences during the last week?”), three measure locus of control (e.g., “To what extent do you feel internally hurried during conversation this past week with the following audiences?”) and three measure avoidance (e.g., How often did you change words during the last week when you thought you might get stuck with the following audiences?”). Each item was scored on a 1–9 Likert scale for close friend, authority figure, and telephone situation. Higher scores reflected a higher stuttering severity. The current Cronbach’s alpha was 0.92).

### 2.4. Statistical Analysis

Data analysis was performed using SPSS^®^ version 28.0 (IBM Corporation, Armonk, NY, USA) for the Apple Mac^®^.

First, we compared sociodemographic information between AWS and AWNS with a t-test and a series of X^2^-tests.

To answer the first hypothesis (AWS would report more sleep complaints compared to AWNS), a multivariate ANOVA was performed with the factor group (AWS; AWNS) and the dimensions of the PSQI as dependent variables. Covariates were age and social anxiety. For F-tests, effect sizes were reported as partial eta-squared [ηp^2^] and interpreted as follows: trivial [T] = η_p_^2^ ≤ 0.019; small [S] = 0.020 ≤ η_p_^2^ ≤ 0.059; medium [M] = 0.06 ≤ η_p_^2^ ≤ 0.139, or large [L] = η_p_^2^ ≥ 0.14. Please note that we focused on effect sizes and not on *p*-values when interpreting the results, as *p*-values might become statistically significant due to the sample size [58,59].

To answer the second hypothesis (poor sleep is related to social anxiety), we performed a series of Pearson’s correlations between scores for sleep disturbances and social anxiety.

To answer the third hypothesis (poor sleep is related to higher stuttering), we performed a series of Pearson’s correlations between scores for stuttering and social anxiety.

To answer the first research question, two partial correlations were performed: a Pearson’s correlation between sleep disturbances and stuttering severity, controlling for social anxiety, and a Pearson’s correlation between sleep disturbances and social anxiety, controlling for stuttering severity.

Cut-off points for Pearson’s correlations were [60,61]: 0.11 ≤ r ≤ 0.30 = small; 0.31 ≤ r ≤ 0.49 = medium; r ≥ 0.50 = large.

To answer the second research question among AWS, two statistical approaches were applied: First, we performed a multiple regression analysis with sleep disturbances as dependent dimension and social anxiety, stuttering severity, and age as predictors. Preliminary conditions to run multiple regressions were met [62,63]: N = 110 > 100; predictors explained the dependent variables (R = 0.345, R^2^ = 0.119); the number of predictors was not 10 times larger than the sample size: 10 × 3 = 30 < N (110), and the Durbin–Watson coefficient was 1.54, indicating that the residuals of the predictors were independent. Furthermore, the variance inflation factors (VIF) were between 1.05 and 1.16; while there are no strict cut-off values to report the risk of multicollinearity, VIF < 1 and VIF > 10 indicate multicollinearity [62,63]. Second, to test for possible interaction effects within the multiple regression model, we followed Aiken and West [64], who proposed to multiply the residuals of the independent factors and to model this product as a function of the categories of one factor. Specifically, we multiplied the residuals of the factors social anxiety and stuttering severity; next, the product was entered into the multiple regression model as a further factor. Last, to understand the nature and direction of the interaction, three correlations were performed with the dimensions sleep disturbances and stuttering severity, mediated by the category of social anxiety (no social anxiety; moderate social anxiety: high social anxiety).

The level of significance was set at *p* < 0.05.

### 2.5. Ethical Consideration

Written informed consent was obtained from all participants. The study was approved by the Ethics Committee of the KUMS (IR.KUMS.REC.1398.972).

## 3. Results

### 3.1. Sociodemographic Findings

Table 1 reports the descriptive and inferential statistical indices of sociodemographic information between AWS and AWNS. While the two groups did not differ with regard to age, socioeconomic status, highest educational degrees, or job position, AWS were primarily males and unmarried.

### 3.2. Subjective Sleep Quality among Adults with (AWS) and without (AWNS) Stuttering

Table 2 reports the descriptive and inferential statistical indices of the total PSQI score and its seven subscales between AWS and AWNS, controlling for age and social anxiety.

As shown in Table 2, compared to AWNS, AWS reported a shorter sleep duration, a lower sleep efficiency, a higher use of sleep medications (significant *p*-values and medium effect sizes), and an overall higher score for sleep disturbances (significant *p*-value and large effect size). Next, while *p*-values were significant for subjective sleep quality, sleep disturbances, and daytime functioning, whenever controlling for age and social anxiety, their effect sizes were trivial or small. For sleep latency, the *p*-value was not significant and the effect size was trivial.

### 3.3. Associations between Subjective Sleep Disturbances and Social Anxiety among Adults with and without Stuttering

Table 3 reports the Pearson’s correlation coefficients between dimensions of subjective sleep disturbances and social anxiety among AWS and AWNS.

For the whole sample, higher scores for social anxiety were associated with higher scores for subjective sleep disturbances, longer sleep latency, a shorter sleep duration, higher sleep disturbances, more use of sleep medications, a higher daytime dysfunctioning and a higher sleep disturbances score. While *p*-values for correlation coefficients were statistically significant, their effect sizes were small.

Among AWS, all correlation coefficients were small. Among AWNS, three correlation coefficients were statistically significant (significant *p*-values), but their effect sizes were small.

### 3.4. Sleep Disturbances and Stuttering Severity among Adults Who Stutter (AWS)

Table 4 reports the Pearson’s correlation coefficients between scores for subjective sleep disturbances and dimensions of stuttering among AWS.

A higher avoidance was associated with a lower subjective sleep quality. Higher sleep disturbances were associated with a lower locus of control, a higher avoidance, and a higher total stuttering severity score. While four correlation coefficients were statistically significant (significant *p*-values), their effect sizes were small.

### 3.5. Partial Correlations between Sleep Disturbances and Stuttering Severity, When Controlling for Social Anxiety, and Partial Correlations between Sleep Disturbances and Social Anxiety, When Controlling for Stuttering Severity among Adults Who Stutter (AWS)

Table 5 reports the Pearson’s correlations coefficients between sleep disturbances and stuttering severity, without and with controlling for social anxiety, and the Pearson’s correlation coefficients between sleep disturbances and social anxiety, without and with controlling for stuttering severity, and always among adults who stutter (AWS).

The correlation coefficient between sleep disturbances and stuttering severity decreased when controlling for social anxiety. The correlation coefficient between sleep disturbances and social anxiety decreased when controlling for stuttering severity.

### 3.6. Predicting Sleep Disturbances among Adults Who Stutter (AWS)

To predict sleep disturbances among adults who stutter (AWS), a multiple regression analysis was performed with sleep disturbances as the dependent variable and stuttering severity, age, social anxiety and the stuttering severity x social anxiety-interaction as possible predictors. Table 6 reports the results of the multiple regression. Older age and higher stuttering severity predicted higher sleep disturbances, while neither social anxiety nor the stuttering severity × social anxiety-interaction had predictive statistical significance.

However, when social anxiety categories (no social anxiety; moderate social anxiety; high social anxiety) were introduced as independent categories and thus as a mediator in the associations between sleep disturbances and stuttering severity, it turned out that the strengths of association between stuttering severity and sleep disturbances were small (r < 0.11) among those AWS with low and moderate social anxiety, while the correlation coefficient between stuttering severity and sleep disturbances was increased among AWS with high social anxiety (r = 0.27).

## 4. Discussion

The aims of the present study were to compare subjective sleep disturbances among adults who stutter (AWS) and adults with no stuttering (AWNS), to associate subjective sleep disturbances with social anxiety and stuttering severity, and to identify predictors to define subjective sleep disturbances among adults who stutter. The key findings of the study were that compared to adults with no stuttering (AWNS), adults who stutter (AWS) reported more impaired sleep disturbances, even when controlling for age and social anxiety. Results from partial correlations showed that social anxiety moderated the associations between sleep disturbances and stuttering severity, while stuttering severity also moderated the association between sleep disturbances and social anxiety. Last, for AWS, higher stuttering severity scores and older age predicted higher sleep disturbances, while social anxiety had no statistical significance. However, results from the mediating model showed that the association between higher stuttering severity and higher sleep disturbances was only significant among those AWS reporting higher scores for social anxiety. In our opinion, the present results add to the current literature in an important way, in that in line with results from children and adolescents who stutter, adults who stutter also self-reported more impaired sleep. Furthermore, from the regression model it turned out that, contrary to expectations, social anxiety scores were unrelated to sleep disturbances, while a higher stuttering severity and older age predicted higher sleep disturbances. However, the association between stuttering severity and sleep disturbances was largest among those AWS in the category of high social anxiety.

Three hypotheses and two research questions were formulated, and each of them is now considered in turn.

With the first hypothesis we expected that compared to adults who do not stutter (AWNS), adults who stutter would (AWS) report higher sleep disturbances, and data did partially confirm this. As shown in Table 2, AWS reported a shorter sleep duration, a lower sleep efficiency, a higher use of sleep medications (medium effect sizes), and an overall higher score for subjective sleep disturbances (large effect size), while effect sizes were trivial to small for subjective sleep quality, sleep latency, sleep disturbances, and daytime functioning, always when controlling for age and social anxiety. The novelty of the results is that to the best of our knowledge, this is the very first study to show higher sleep disturbances not only among children and adolescents who stutter [42,43,44], but also among adults who stutter.

With the second hypothesis we assumed that higher scores for sleep disturbances would be associated with higher social anxiety. This assumption was based on studies with samples with no stuttering [37,38,39,40,41]. However, against expectations, the data did not confirm this assumption (see Table 3), in that correlation coefficients were small, and in that the significant *p*-values appeared to be due to the larger sample size. However, in support of the hypothesis, we may also acknowledge that the correlation coefficients went to the expected directions. As such, though to a small statistical extent, social anxiety and higher sleep disturbances were associated. Furthermore, the correlation coefficient of r = 0.256 indicates that 6.5% (r^2^ = 0.065) of the variance of social anxiety explained the variance of sleep disturbances, while, complementarily, 93,5% of the variance of sleep disturbances could not be explained by social anxiety. As such, latent and unassessed dimensions might have had a higher power on the prediction of poor sleep among AWS.

With the third hypothesis we assumed that higher scores for stuttering severity would predict higher scores for sleep disturbances, and data did to some extent confirm this, as shown in Table 4.

Given that the present study was the first to investigate sleep disturbances and stuttering severity among adults who stutter, the theoretical basis was as follows: given that stuttering was associated with social anxiety [5,14,15,16,17,18,19,20,21,22,23], and given that poor sleep was associated with higher anxiety scores [37,38,39,40,41], we expected that higher scores for stuttering severity would be associated with higher scores for sleep disturbances. However, the correlation coefficient of r = 0.246 indicated that 6% (R^2^ = 0.060) of the variance of stuttering severity explained the variance of sleep disturbances, while, complementarily, 94% of the variance of sleep disturbances could not be explained by stuttering severity. While results from the partial correlation computations (see Table 5) showed that social anxiety had a contribution as moderator, irregular sleep habits and sleep deprivation were identified as further possible factors [65,66]. However, the present result is the first one that examined the association between the severity of sleep disturbance and the severity of stuttering.

Next, sleep disturbances can interrupt emotional, cognitive, linguistic, and motor functions. Studies have confirmed the negative effects of poor sleep quality and sleep disorders on working memory function [67,68,69], though working memory is also a very important component of developmental stuttering [70,71,72]. In addition, evidence indicated the deficit in short-term memory, inhibition, and attention among persons with stuttering [73], functions which are negatively impacted by sleep disorders [74,75,76].

With the first research question we asked if, among AWS, social anxiety might moderate the association between sleep disturbances and stuttering severity; complementarily, we also asked if stuttering severity might moderate the association between sleep disturbances and social anxiety. As shown in Table 5, both social anxiety and stuttering severity moderated the specific association, though the influence was again small.

With the second research question, we asked which dimensions could predict scores for sleep disturbances among adults who stutter. As shown in Table 6, and in line with previous results of the present study, older age and higher stuttering severity predicted higher sleep disturbances, while social anxiety did not reach statistical significance. However, when running a mediator model with stuttering severity x social anxiety-interaction as a further predictor, it turned out that the interaction predictor was not significant, but the degree of association between stuttering severity and sleep disturbances was highest among those AWS belonging to the category of high social anxiety.

Despite the intriguing results, the following limitations should be considered. First, strictly taken, the cross-sectional study design does not allow conclusive causal directions: while it is conceivable that poor sleep unfavorably impacts speech performance and social behavior, subjectively perceived speech performance and social behavior might also unfavorably impact on sleep quality, though bi-directional processes might best describe such types of associations over time. Second and relatedly, the decision was to run a mediation model. In this view, while a regression model pre-defines predictors and dependent variables (thus, causal relationships), such regression models are helpful to answer the research questions. Third, objective sleep assessments might have provided further information on sleep patterns, as subjective sleep assessments might be biased via current mood states. Fourth, latent and unassessed psychological dimensions such as symptoms of depression, self-efficacy, subjectively perceived social support or social rejection might have biased two or more dimensions in the same or opposite directions. Fifth, the study was performed during the COVID-19 pandemic and its related social restrictions. There are no consistent and straightforward data which showed that the COVID-19 pandemic and its related social restrictions negatively impacted the sleep of the general population: while some studies reported a considerable change in sleep patterns among the general population during the pandemic (e.g., [77,78]), other studies showed exactly the opposite [79]. Furthermore, it turned out that, above all, the very first studies on the impact of the COVID-19 pandemic and its related social restrictions were not consistently performed in a proper fashion [80]. As such, we recognize that the COVID-19 pandemic and its related social restrictions might have biased the present pattern of results, though the direction of such a bias remains elusive.

## 5. Conclusions

Compared to adults with no stuttering (AWNS), adults with stuttering (AWS) self-reported some, but not consistent, sleep impairments. Such sleep disturbances among AWS were associated with older age and higher stuttering severity. but not with social anxiety, in general, though, specifically, poor sleep and higher stuttering severity was observed among those AWS scoring high on social anxiety. Given this, adults who stutter should be routinely asked about their sleep quality.

## Figures and Tables

**Table 1 jcm-12-00161-t001:** Demographic characteristics among adults with (AWS) and with no stuttering (AWNS).

		AWS (n = 110)	AWNS (n = 162)		*p*-Value
		M (SD)	M (SD)		
Age (Years)		28.26 (6.58)	29.04 (9.43)	t (270) = 0.76	0.22
		n (%)	n (%)	*X^2^*	
**Gender (%)**	Male	80 (72.70)	79 (48.80)	15.49	<0.0.1
	Female	30 (27.30)	83 (51.20)		
**Economic status**	High income	17 (15.50)	35 (21.60)	2.05	0.36
	Moderate	72 (65.50)	103 (63.60)		
	Low income	21 (19.10)	24 (14.80)		
**Education**	High school	20 (18.20)	41 (25.30)	2.56	0.28
	Undergraduate	67 (60.90)	84 (59.10)		
	Postgraduate	23 (20.90)	37 (22.80)		
**Job**	Unemployed	35 (31.80)	71 (43.80)	5.39	0.07
	Employed	36 (32.70)	52 (32.10)		
	Self-employed	39 (35.50)	39 (24.10)		
**Marriage status**	Married	22 (20.00)	62 (38.30)	13.02	<0.01
	Unmarried	88 (80.00)	97 (59.90)		
	Divorced	0	3 (1.90)		

AWS: adults who stutter. AWNS: adults who not stutter.

**Table 2 jcm-12-00161-t002:** Descriptive and inferential statistical overview of sleep disturbances, separately for adults who stutter (AWS) and adults who do not stutter (AWNS).

Group	AWS	AWNS	F	Effect Size
N	110	162		
Pittsburgh Sleep Quality Index	M (SD)	M (SD)		
Subjective sleep quality	1.21 (0.79)	0.93 (0.66)	7.13 **	0.026 [T]
Sleep latency	2.10 (0.78)	1.92 (0.82)	1.84	0.007 [T]
Sleep duration	1.01 (0.92)	0.43 (0.71)	31.07 ***	0.104 [M]
Sleep efficiency	0.85 (1.14)	0.25 (0.63)	28.94 ***	0.097 [M]
Sleep disturbances	1.80 (0.55)	0.16 (0.48)	6.48 *	0.024 [S]
Use of sleep medication	0.62 (0.94)	0.16 (0.48)	24.29 ***	0.083 [M]
Daytime dysfunction	1.66 (0.91)	1.22 (0.89)	11.83 ***	0.042 [S]
Total PSQI score	9.26 (3.43)	6.52 (2.48)	49.49 ***	0.156 [L]

AWS: adults who stutter. AWNS: adults who not stutter. * = *p* < 0.05; ** = *p* < 0.01; *** = *p* < 0.001. [T] = trivial effect size; [S] = small effect size; [M] = medium effect size; [L] = large effect size.

**Table 3 jcm-12-00161-t003:** Pearson’s correlation coefficient scores for subjective sleep disturbances and social anxiety for the whole sample, and separately for AWS and AWNS.

	Whole Sample	AWS	AWNS
N	272	110	162
Pittsburgh Sleep Quality Index	Social anxiety	Social anxiety	Social anxiety
	r	r	r
Subjective sleep quality	0.164 **	0.122	0.139
Sleep latency	0.126 *	0.98	0.116
Sleep duration	0.131 *	0.028	0.112
Sleep efficiency	0.100	0.019	0.063
Sleep disturbances	0.193 **	0.134	0.188 *
Use of sleep medication	0.147 **	0.073	0.125
Daytime dysfunction	0.193 **	0.139	0.164 *
Total PSQI score	0.256 ***	0.142	0.248 **

AWS: adults who stutter; AWNS: adults who do not stutter; * = *p* < 0.05; ** = *p* < 0.01; *** = *p* < 0.001.

**Table 4 jcm-12-00161-t004:** Pearson’s correlation coefficient scores for subjective sleep disturbances and stuttering severity among AWS (N = 110).

	Stuttering			
	Severity	Locus of Control	Avoidance	Total Score
Pittsburgh Sleep Quality Index	R	r	r	r
Subjective sleep quality	0.035	0.138	0.195 *	0.164
Sleep latency	−0.065	0.000	−0.054	−0.037
Sleep duration	0.105	−0.006	−0.049	−0.0004
Sleep efficiency	0.095	−0.066	−0.049	−0.035
Sleep disturbances	0.170	0.211 *	0.235 *	0.246 **
Use of sleep medication	0.104	−0.032	0.060	0.034
Daytime dysfunction	0.053	0.113	0.139	0.130
Total PSQI score	0.123	0.064	0.094	0.100

AWS: adults who stutter; * = *p* < 0.05; ** = *p* < 0.01.

**Table 5 jcm-12-00161-t005:** Pearson’s correlations coefficients between sleep disturbances and stuttering severity, without and with controlling for social anxiety, and the Pearson’s correlation coefficients between sleep disturbances and social anxiety, without and with controlling for stuttering severity, always among AWS.

	Stuttering Severity	Social Anxiety
Sleep disturbances	r = 0.100	r = 0.142
Controlling for social anxiety	r_p_ = 0.051	
Controlling for stuttering severity		r_p_ = 0.114

**Table 6 jcm-12-00161-t006:** Multiple regression model to predict sleep disturbances by age, social anxiety, and stuttering severity among adults who stutter.

		Non-Standardized Coefficients		Standardized Coefficient						
Dimension	Variable	Coefficient Beta	Standard Error	Beta	t	*p*	R	R^2^	R^2^ Corr	Durbin-Watson Statistics
Sleep disturbances	Intercept	3.923	2.109	-	2.861	0.021	0.252	0.063	0.028	1.587
	Social anxiety	0.123	0.092	0.137	1.343	0.182				
	Stuttering severity	0.003	0.015	0.229	2.012	0.032				
	Social anxiety x stuttering severity	0.001	0.0004	0.016	0.171	0.984				
	Age	0.016	0.050	0.203	2.125	0.036				

## Data Availability

Data files and analysis are available upon request of confirmed experts in in the field and upon clear-cut formulations of hypotheses, and upon the statement of the confidential and secure storage and handling of the data.

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
