# Peer review of "Sleep Problems, Social Anxiety and Stuttering Severity in Adults Who Do and Adults Who Do Not Stutter"

_jcm, 2022, doi:10.3390/jcm12010161_

Round 1

Reviewer 1 Report

This manuscript addresses an interesting and important topic. It is well written. The background is clearly presented in the Introducition. A large, carefully selected sample was included. Methods are timely and are fully  appropriate. Novel interesting  results are provided in text and tables. Discussion is full appropriate. Limitations are described.

I raise only few critical comments.

Is the Iranian stuttering association  a society of exüerts or a support group? Please clarify. 

In the abstract "drug use" is mentionned. Please clarfy already here sleeping pills.

line 133 - please replace dot following 61] by hyphen.

please  correct in line 360 the misspelled word "unfavourably". 

Author Response

We thank Reviewer #1 for their helpful and encouraging comments, which helped us to improve the quality of the revision. 

Please find the detailed point-by-point-response attached as a separate file.

Thank you once again. 

Reviewer 2 Report

The authors examined sleep quality, social anxiety, and stuttering severity in adults with and without stuttering. The findings in the adult population are novel and potentially clinically relevant, but I have the following comments to help improve this manuscript:

Major concerns:

1. Line 131-134 (also line 330-333). The authors stated that because stuttering is associated with social anxiety, and also because poor sleep is associated with social anxiety, they predicted that stuttering severity would be associated with sleep quality. Here, the authors seem to suggest that social anxiety mediates the relationship between stuttering severity and sleep quality. 

However, my takeaway from the introduction is that stuttering affects social anxiety (line 65-87) and that sleep affects stuttering severity (line 93-101). In this case, the relationship between sleep and social anxiety should be mediated by stuttering. I suggest the authors clarify the mechanisms between the three constructs in the introduction and discussion. And if the aim is to untangle the associations between the three constructs (line 137-139), more sophisticated statistical analyses (e.g., mediation analysis) may be helpful. 

2. Line 261-262. There are significant results presented in Table 3, but the authors stated no significant findings in line 261-262, line 293, and line 317-320. If the authors believe that these correlations are too small to be considered meaningful, then the criteria for determining meaningful correlations should be established beforehand and the same criteria should be applied throughout (e.g., r=0.256 in Table 3 is considered no correlation, but r=0.246 in Table 4 is considered a correlation).

Minor concerns:

1. Line 37-38. Though the results for subjective sleep quality and sleep disturbances, and daytime dysfunction yielded trivia/small effects, they are statistically significant. So it is incorrect to state that "no statistically significant mean differences were observed". 

2. Line 68, the acronym AWNS was not defined.

3. Line 123-125. Both the second and the third aims include associated self-rated sleep quality with dimensions of stuttering severity. They seem repetitive.

4. Results section: it would be great if the authors can report the age distributions of both groups, and test whether age differs between groups.

5. No novel results should be introduced in the discussion section (line 312-313). The authors should report these statistics in the results section and test whether there are differences between groups.

Author Response

We thank Reviewer #2 for their helpful and encouraging comments, which helped us to improve the quality of the revision. 

Please find the detailed point-by-point-response attached as a separate file.

Thank you once again. 

Round 2

Reviewer 2 Report

In this revised manuscript, the authors have addressed most of my previous concerns. However, the partial correlation analyses do not satisfactorily address my first comment in the previous review. 

Partial correlations can be helpful for testing mediation, but not moderation. Moderation means the relationship between factor X and factor Y depends on the level of factor M. It should be tested by an interaction effect. 

My initial intention was to ask the authors to clarify the relationships between sleep, stuttering, and social anxiety. The nature of this study (non-RCT) does not allow inference of causal relationships, so the authors need to explain the underlying mechanisms based on existing theories/evidence. 

What causes confusion is that in the current background (line 97-102), the authors explained that sleep affects stuttering by affecting cognition and motor control. But when stating the hypotheses (line 132-135) and explaining the mechanisms (line 363-366), the authors stated that sleep is associated with stuttering via social anxiety. I would suggest the authors be consistent when explaining the mechanisms. The partial correlation analyses are not very helpful and could be dropped. Mediation analyses would be helpful, if the authors would like to test whether sleep is associated with stuttering via social anxiety. 

Author Response

Again, we thank Reviewer #2 for helping us to improve the quality of the present manuscript. We did highly appreciate the Reviewer's expertise and suggestions. 
